# Massively Parallel Profiling of HIV-1 Resistance to the Fusion Inhibitor Enfuvirtide

**DOI:** 10.3390/v11050439

**Published:** 2019-05-15

**Authors:** Adam S. Dingens, Dana Arenz, Julie Overbaugh, Jesse D. Bloom

**Affiliations:** 1Basic Sciences and Computational Biology, Fred Hutchinson Cancer Research Center, Seattle, WA 98109, USA; 2Human Biology, Fred Hutchinson Cancer Research Center, Seattle, WA 98109, USA; darenz@fredhutch.org (D.A.); joverbau@fredhutch.org (J.O.); 3Howard Hughes Medical Institute, Seattle, WA 98109, USA

**Keywords:** HIV-1 drug resistance, enfuvirtide, T-20, fusion inhibitor, HIV-1 envelope, deep mutational scanning

## Abstract

Identifying drug resistance mutations is important for the clinical use of antivirals and can help define both a drug’s mechanism of action and the mechanistic basis of resistance. Resistance mutations are often identified one-at-a-time by studying viral evolution within treated patients or during viral growth in the presence of a drug in cell culture. Such approaches have previously mapped resistance to enfuvirtide, the only clinically approved HIV-1 fusion inhibitor, to enfuvirtide’s binding site in the N-terminal heptad repeat (NHR) of the Envelope (Env) transmembrane domain as well as a limited number of allosteric sites. Here, we sought to better delineate the genotypic determinants of resistance throughout Env. We used deep mutational scanning to quantify the effect of all single-amino-acid mutations to the subtype A BG505 Env on resistance to enfuvirtide. We identified both previously characterized and numerous novel resistance mutations in the NHR. Additional resistance mutations clustered in other regions of Env conformational intermediates, suggesting they may act during different fusion steps by altering fusion kinetics and/or exposure of the enfuvirtide binding site. This complete map of resistance sheds light on the diverse mechanisms of enfuvirtide resistance and highlights the utility of using deep mutational scanning to comprehensively map potential drug resistance mutations.

## 1. Introduction

Antiretroviral drug therapy has reduced the global burden of HIV/AIDS. However, HIV-1’s exceptional evolutionary capacity enables the virus to evolve resistance, eroding the therapeutic efficacy of drug regimens. Identifying drug resistance mutations is important to the clinical management of HIV-1/AIDS, as well as in the development of new drugs that are less prone to resistance. In most cases, resistance mutations are defined when they arise in treated patients and in a one-by-one basis in viral cell culture experiments. Methods to comprehensively identify resistance mutations are lacking.

We chose enfuvirtide, the first and only clinically approved fusion inhibitor, as a model drug to test a high-throughput approach to map drug resistance. Enfuvirtide (also known as T20 and Fuzeon), is a 36 amino-acid peptide derived from the C-terminal heptad repeat (CHR) domain of gp41 that inhibits viral entry into cells [1,2]. During fusion, receptor and co-receptor binding each induce conformational rearrangements to Env resulting in the formation of an extended pre-hairpin intermediate conformation [3]. Subsequently, Env’s CHR collapses onto the coiled-coil NHR structure of the pre-hairpin intermediate, forming the six-helix bundle that pulls the cell and viral membranes close enough for fusion [3]. Enfuvirtide mimics the CHR and binds to the NHR in the pre-hairpin intermediate, thereby inhibiting fusion by blocking the formation of the six-helix bundle [3,4].

Defining enfuvirtide resistance mutations can help elucidate both the mechanisms of resistance and the conformational dynamics during Env’s fusion process [4,5,6,7,8]. Understanding the mechanistic basis of enfuvirtide resistance has also aided in developing new fusion inhibitors less prone to viral resistance [9,10,11,12,13]. While enfuvirtide is rarely used as a part of current clinical regimens, resistance can be of clinical importance. Due to enfuvirtide’s high cost and need for twice-daily subcutaneous injections, it is only used as part of a “salvage therapy” for highly treatment-experienced patients with multidrug resistance. Such patients need to be carefully monitored for resistance before and during therapy [14], which requires knowing the viral genotypic determinants of enfuvirtide resistance. 

Enfuvirtide resistance can occur directly via mutations in the drug’s binding site that disrupt enfuvirtide binding, or indirectly via mutations allosteric to the binding site that alter sensitivity in more complex ways [15]. Studies that analyzed enfuvirtide resistance in cell culture and in vivo during therapy have identified resistance mutations at sites 547−556 in enfuvirtide’s binding site in the NHR [15,16,17,18,19,20,21,22,23,24]. Outside of the enfuvirtide’s binding site, mutations in the CHR may be able to play a compensatory role in increasing enfuvirtide resistance [19,25]. Additionally, gp120 mutations in the V3 loop and co-receptor binding site have also been shown to affect enfuvirtide sensitivity, likely by altering co-receptor tropism, co-receptor affinity, or fusion kinetics [7,26,27,28]. 

Much of the prior work on enfuvirtide sensitivity and resistance has focused on subtype B viruses [17,18]. Sequence variability within enfuvirtide’s binding site differs across subtypes, but a limited number of studies have suggested there are not substantial differences in enfuvirtide sensitivity or resistance mutations between subtypes [29,30,31,32,33].

Here, we more completely map resistance mutations throughout Env. We used deep mutational scanning to quantify how enfuvirtide resistance is affected by all mutations to Env compatible with viral replication in the background of the HIV-1 subtype A virus BG505. 

## 2. Materials and Methods

### 2.1. Generation of Env Mutant Virus Libraries

We used Env mutant virus libraries from the subtype A, transmitted variant BG505 to map enfuvirtide resistance. We used this strain because relatively little is known about enfuvirtide resistance in subtype A viruses, and because BG505 has been widely used in structural studies [34,35]. Briefly, we independently generated triplicate proviral DNA libraries, encoding codon-level mutations to sites 31-702 (HXB2 numbering is used throughout the manuscript) of BG505.W6M.C2.T332N *env*. These libraries contain (670 mutagenized sites) × (19 amino acid mutations) = 12,730 possible amino-acid mutations. We then produced viruses from these mutant DNA libraries and passaged them in SupT1.CCR5 cells to select for functional viruses, resulting in mutant virus libraries that encode all functionally tolerated mutations to Env [34]. 

### 2.2. Resistance Profiling

To identify resistance mutations, we incubated the mutant virus libraries with or without enfuvirtide, infected cells, and then identified the mutant viruses that were enriched upon drug selection using deep sequencing. This approach is similar to the mutational antigenic profiling process we have previously used to map antibody escape [36,37,38]. Briefly, 5 × 10^5^ to 1 × 10^6^ infectious units of three independent mutant virus libraries were incubated in the presence of 8 μg/mL of enfuvirtide, then infected into 1 × 10^6^ SupT1.CCR5 cells in R10 (RPMI with 10% FBS, 1% 200 mM L-glutamine, and 100 units/mL of penicillin and streptomycin), containing 100 μg/mL DEAE-dextran. Three hours post infection, cells were resuspended in 1 mL R10 without DEAE-dextran. At 12 hours post infection, non-integrated viral cDNA was isolated using a miniprep. As mock-selected controls, each mutant virus library was infected into cells without enfuvirtide selection. Selected and mock-selected viral cDNA was then sequenced with a barcoded subamplicon sequencing approach as previously described [34], which introduces unique molecular identifiers used to correct sequencing errors. The amount of virus library that entered cells was quantified via qPCR [37].

### 2.3. Analysis of Deep Sequencing Data

We used dms_tools2 version 2.3.0 (https://jbloomlab.github.io/dms_tools2/) to analyze the deep sequencing data [39]. Differential selection has been previously described [40] and is documented at https://jbloomlab.github.io/dms_tools2/diffsel.html. Briefly, the differential selection for each mutation was quantified as the logarithm of the mutation’s enrichment in the enfuvirtide-selected mutant virus library relative to the non-selected control library. Sequencing of wildtype DNA plasmid was used as the error control while calculating differential selection. Throughout this manuscript, we focused on the median differential selection metrics of biological triplicate experiments.

### 2.4. Data Availability and Source Code

The computational analysis is provided as Appendix A and at https://github.com/jbloomlab/EnfuvirtideResistance. Differential selection measurements are provided as Appendix A. Illumina reads were deposited into the NCBI SRA as SRR8097918- SRR8097920. 

### 2.5. TZM-BL Inhibition Assays

Individual mutations were introduced into BG505.T332N Env, and pseudoviruses were generated and tested in TZM-bl inhibition assays as previously described [36]. Each mutant was tested independently twice, and each experiment was performed in duplicate. Inhibition curves were fit independently for each experimental replicate using 4-parameter nonlinear regression. Inhibition of wildtype pseudovirus was performed on each plate to reduce noise. The fold change in IC_50_ relative to wildtype and the maximum percent inhibition for each mutant were computed for each experimental replicate, then averaged across replicates.

## 3. Results

We have previously generated triplicate mutant virus libraries that contain all single-amino-acid mutations to the ecto and transmembrane domain of BG505.W6M.C2.T332N Env compatible with viral replication in cell culture [34]. We selected these triplicate mutant virus libraries with a highly selective enfuvirtide concentration (8 μg/mL), which resulted in just 0.15% to 0.78% of the mutant virus libraries surviving selection (Appendix A). We infected SupT1.CCR5 cells and deep sequenced the env genes of viruses that entered cells. Lastly, we calculated the enrichment of each mutation in the drug-selected condition relative to a non-selected mutant virus library, termed the differential selection [40]. These experimental conditions, including the enfuvirtide concentration and cell line, were optimized for high-throughput screening of potential drug resistance mutations rather than to mimic the far more complex in vivo therapeutic conditions. 

Selection of resistance mutations was highly reproducible across biological triplicates (Appendix A). Most resistance mutations were found in the NHR domain (Figure 1, Appendix A) and included many well-characterized mutations on the IAS–USA drug resistance mutations list [41]. For example, S553T, a known resistance mutation, was the largest effect size mutation at site 553 in our experiments. Similarly, V549E and V549A, both known resistance mutations, were the largest effect size mutations at site 549. However, V549M was not enriched in our experiments despite being associated with resistance in other strains. Similarly, the known I548V resistance mutation was not enriched, but numerous other mutations at site 548 were strongly enriched. Of note, most prior resistance work has been performed in clade B viruses, while we used a clade A virus; we discuss the potential differences in resistance across strains in more detail in the Discussion. 

Overall, this data suggests there are many more possible resistance mutations in the NHR region than those that have been previously characterized. For example, while the known resistance mutation Q551H was enriched in our experiments, many additional amino acids at this site also increased enfuvirtide resistance, with 12 other mutations at this site having a larger effect than Q551H (Figure 1). Similarly, while the common resistance mutations N554D/K were enriched, other mutations to 554 had even larger effect sizes. Overall, we identified a much larger set of mutations compared to the IAS-USA drug resistance mutations list, including many mutations to additional sites in NHR (such as sites 552, 556, 557, and 560), that were enriched upon enfuvirtide selection (Figure 1). 

This large set of potential NHR resistance mutations suggests resistance in this region may occur via different mechanisms. Site 551, where many different mutations confer resistance, directly interacts with enfuvirtide [4], suggesting this may directly interfere with enfuvirtide binding. Surprisingly, the side chains at the next largest effect-size sites (548 and 552) face the NHR trimer center and interact with other NHR α-helices (Figure 2A) [4,8]. Resistance at these sites occurred primarily via positively charged or bulky amino acids (Figure 1B). The next two largest effect sites in this region, 554 and 556, interact with enfuvirtide (Figure 2A). 

There was also modest, but reproducible enrichment of mutations at other Env sites outside of the NHR domain. One such mutation was P76Y, which interacts with NHR sites L555 and L556 in the prefusion conformation (Figure 2B). Other potential resistance mutations occurred at sites 424–436 in the β20/β21 strand of C4, as well as sites 119, 121, and 207 in the V1/V2 stem. While the V1/V2 stem is distant from β20/β21 in the prefusion Env conformation, it shifts upon CD4 binding to form the 4-stranded bridging sheet along with the β20/β21 strand, creating the portion of the co-receptor binding site that interacts with the N-terminus of CCR5 [42]. This cluster of potential resistance mutations extended to site 111 present below the bridging sheet in Env’s CD4- and CCR5-bound state.

To validate that our high-throughput mapping accurately identifies mutations that increase resistance to enfuvirtide in cell culture, we generated and tested individual BG505 Env pseudoviruses bearing single mutations for enfuvirtide sensitivity. We selected both previously characterized and novel resistance mutations from each of the clusters of resistance mutations. The V549E and Q552R mutations increased resistance, shifting the IC_50_ by >150-fold (Figure 3). Other mutations that were modestly enriched (P76Y, C119R, K121P, and K207L) had little effect on IC_50_ but instead altered the slope and/or decreased the maximal inhibition plateau at the 8 μg/mL enfuvirtide concentration used in resistance profiling (Figure 3), suggesting these mutations may result in a subpopulation of resistant viruses. This agrees with prior work characterizing how enfuvirtide resistance can affect the inhibition curve slope [43]. Notably, both these validation experiments and the resistance profiling itself were performed with a high concentration of infection enhancer (100 μg/mL DEAE-dextran). When the assays were repeated with 10 μg/mL DEAE-dextran, some of the resistance phenotypes were less prominent (Appendix A). 

## 4. Discussion

We have quantified the effect of all single-amino-acid mutations to the extracellular and transmembrane ectodomain of BG505 Env on resistance to the fusion inhibitor enfuvirtide in cell culture. This map of resistance mutations included both previously characterized and numerous novel resistance mutations. The comprehensive aspect of these data defined clusters of mutations that likely alter enfuvirtide sensitivity via different mechanisms and at different steps during fusion.

Even within the NHR, the selected mutations also help elucidate multiple potential mechanisms of resistance. While some NHR mutations may directly disrupt interactions with enfuvirtide (e.g., site 551), others appear to introduce positive charges or bulky amino acids at the center of the NHR coiled-coil (e.g., sites 548 and 552). These mutations may slightly alter the coiled-coil structure to disrupt enfuvirtide binding or favor the intramolecular binding of the CHR domain over binding to enfuvirtide. This hypothesis is supported by a study showing that the Q552R mutation results in an asymmetric six-helix bundle structure with the positively charged 552R residues oriented away from the coiled-coil interface, which alters enfuvirtide’s binding sites [8]. It is possible that this dynamic is influenced by viral subtype, as enfuvirtide is a mimetic of a subtype B CHR domain and differs from BG505 at 7 of 36 amino acids. 

Other resistance mutations likely alter fusion kinetics or accessibility of the enfuvirtide binding site. P76Y may result in additional hydrophobic interactions between site 76 and the NHR sites L555/L556 in the prefusion conformation, but site 76 is not known to interact with the NHR in later Env states that enfuvirtide binds. This interaction could directly limit access of enfuvirtide to the binding site or delay the formation of the enfuvirtide-sensitive intermediate present post CD4-binding. 

We also uncovered many small-effect size resistance mutations in the co-receptor binding site. It has been previously shown that specific mutations to V3 and the bridging sheet can affect enfuvirtide sensitivity by altering CCR5 affinity or fusion kinetics [7,26,27,28]. However, we uncovered additional potential resistance mutations at numerous sites in or near the bridging sheet, including sites in the V1/V2 loop stem and the β20/β21 strand of C4. These mutations cluster in the binding site of CCR5’s N-terminus [42], suggesting they may also increase CCR5 receptor affinity or fusion kinetics, limiting the time in which a CD4-induced enfuvirtide sensitive conformational intermediate is exposed. However, mutations to the β20/β21 strand have also been shown to regulate the transition between “closed” and “open”, CD4-bound-like Env state [44], suggesting these mutations could possibly accelerate the first step of entry. Additionally, there is also evidence of weak interactions between enfuvirtide and the co-receptor binding site on gp120 [45,46]. Resistance mutations in this region could disrupt this potential secondary interaction between enfuvirtide and Env.

The complex mechanisms of enfuvirtide resistance were highlighted further upon validating resistance mutations. While mutations in the NHR shifted the inhibition curve, other tested mutants decreased the maximum inhibition plateau rather than shifting the curve. This suggests that these mutations result in a sub-population of enfuvirtide resistant viruses, possibly by altering Env’s conformation or dynamics. While this is a well characterized mechanism of resistance for neutralizing antibodies [36,47], less data exists on incomplete enfuvirtide inhibition. However, it has been shown that enfuvirtide resistance mutations can alter the inhibition curve slope [43], which may be mechanistically related to the incomplete inhibition observed here. If this phenotype can manifest in vivo and a sub-population of viruses were resistant to enfuvirtide, this could present an opportunity for additional resistance mutations to arise. 

While these data can be considered when evaluating clinical resistance, they also come with some caveats. We examined the effect of single amino acids in a single subtype A Env. Our approach cannot identify compensatory or epistatically interacting resistance mutations, which have been described for enfuvirtide [19,25]. Further, many patients receiving enfuvirtide have subtype B infections, and there could be strain-specific differences in enfuvirtide resistance. For example, we do not identify the G547D and G547S resistance mutations, which were identified by enfuvirtide selection of lab-adapted, subtype B viruses in cell culture [24]. Previous studies show that these mutations are tolerated during viral replication in cell culture in a lab-adapted subtype B Env (LAI), but they are not well-tolerated by the subtype A BG505 Env (Appendix A) [34,35]. Similarly, the known resistance mutation I548V, which was not enriched in our experiments, is better tolerated in LAI than in BG505 (Appendix A). These differences highlight how the balance between enfuvirtide resistance and replicative fitness may depend on the virus strain. Additionally, the resistance phenotypes we map may be specific to our experimental system. Indeed, the phenotype of some of the mutations we validated, particularly the mutations that resulted in incomplete inhibition in a cell culture inhibition assay, were dependent on the DEAE-dextran concentration. This could be related to the well-characterized phenomena that enfuvirtide sensitivity phenotypes commonly measured using cell culture-based experiments do not always correlate with enfuvirtide’s clinical efficacy [17,20,48]. 

In summary, we have used a deep mutational scanning approach to comprehensively map resistance to the HIV fusion inhibitor enfuvirtide. This map of enfuvirtide resistance identifies numerous known and previously uncharacterized mutations, highlighting the utility of using viral deep mutational scanning approaches to evaluate drug resistance. The identification of these mutations may also provide opportunity to better understand the mechanistic bases of resistance; future experiments can confirm these mechanisms proposed here. Additionally, similar studies using subtype B strains may be of use in the clinical monitoring of resistance during therapy or the genotypic prediction of enfuvirtide sensitivity. Lastly, these data highlight how the diverse mechanisms of enfuvirtide resistance may act at different stages throughout Env’s fusion process.

## Figures and Tables

**Figure 1 viruses-11-00439-f001:**
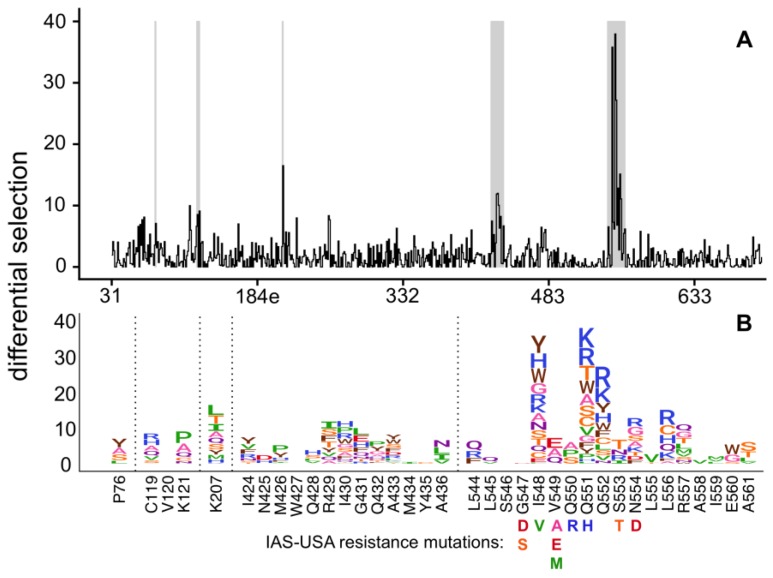
A complete map of enfuvirtide resistance. (**A**) The positive site differential selection is plotted across the mutagenized portion of Env. (**B**) The mutation-level resistance profile for regions of interest (highlighted in grey in (**A**)). See Appendix A for the entire *env*-wide, mutation-level resistance profile. The height of each amino acid is proportional to its differential selection, which is the logarithm of the relative enrichment of that mutation in the enfuvirtide-selected condition relative to the non-selected control. All mutations from the 2017 IAS-USA enfuvirtide resistance mutations list [14] are labeled below their site.

**Figure 2 viruses-11-00439-f002:**
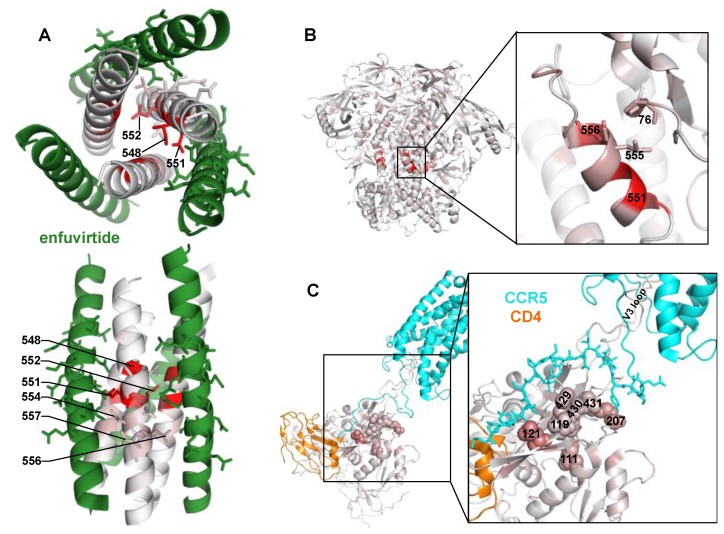
Locations of potential resistance mutations in different Env conformations. (**A**) A structural model of enfuvirtide (green) binding to the N39 NHR coiled-coil trimer (colored white to red according to the positive site differential selection at each site). Sites 547–557 of the NHR, and sites 136-150 of enfuvirtide (using Env numbering) are shown with sticks to highlight side chain positions within the binding interface. PDB:5ZCX. (**B**) The closed, pre-fusion conformation of Env colored white to red according to the maximum mutation differential at each site. PDB:5FYK. (**C**) The CCR5- (cyan) and CD4- (orange) bound gp120 structure, colored white to red according to the maximum mutation differential at each site. The top 2% of resistance sites (*n* = 13 of 670 mutagenized sites, 9 of which are in this gp120 structure) are shown with spheres. Residues 1 to 18 of CCR5 are shown with sticks to indicate bridging sheet interactions. PDB:6MEO.

**Figure 3 viruses-11-00439-f003:**
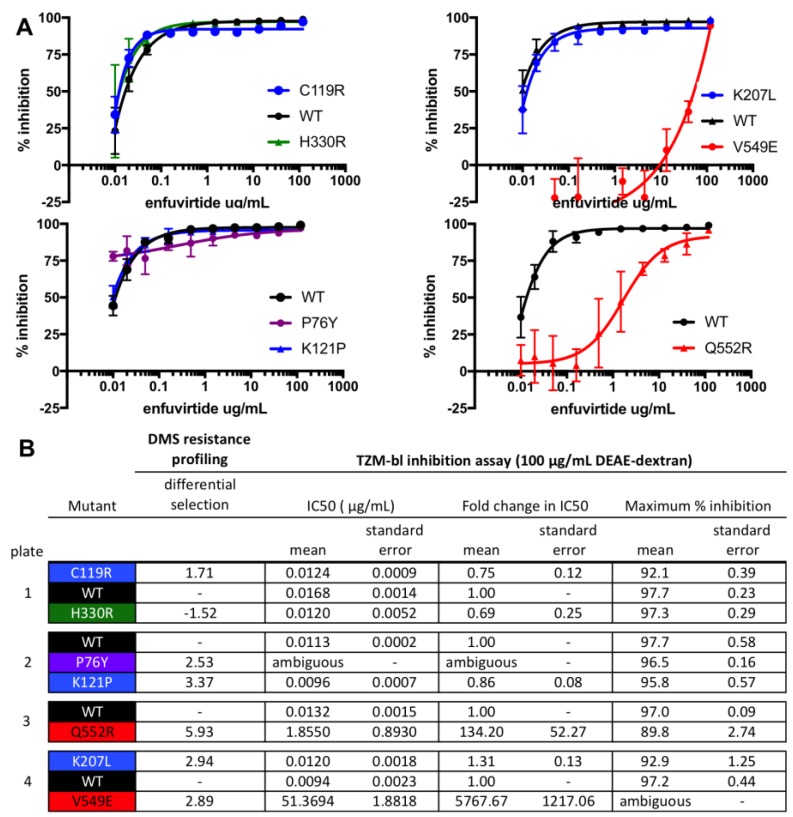
Validation of enfuvirtide resistance mutants using a TZM-bl inhibition assay. TZM-bl inhibition assays were performed in the presence of 100 μg/mL DEAE-dextran, similar to the resistance profiling. (**A**) Inhibition curves are the average of two biological replicates, each performed in duplicate. (**B**) The IC_50_, the fold change in IC_50_ relative to wildtype (WT), and the maximum percent inhibition for each mutant, determined from the fit four-parameter logistic curves. WT virus was run on each plate, and each mutant virus curve was compared to the plate internal WT control. The standard error of the mean is also shown. H330R, which was not enriched in the resistance profiling, was included as a control. In (**A**,**B**), mutant pseudoviruses are colored according to groups (black: WT; green: control mutant not expected to affect enfuvirtide sensitivity; blue: mutants in the V1/V2 Stem/co-receptor binding site; red: mutants in/near NHR binding site).

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
