# Peer review of "Massively Parallel Profiling of HIV-1 Resistance to the Fusion Inhibitor Enfuvirtide"

_viruses, 2019, doi:10.3390/v11050439_

Round 1

Reviewer 1 Report

Summary

The authors use their previously published methods involving deep mutational scanning to quantify the effect of all single amino-acid mutations in a clade A Env on resistance to enfuvirtide. They identified many mutations that were previously identified as resistance mutations in prior clinical and in vitro studies, as well as mutations that have not been previously identified. Most mutations clustered in the NHR, a region known to be important for enfuvirtide binding, but novel mutations in other regions of Env were also identified. The NHR mutations included some, but not all, of the most common enfuvirtide resistance mutations that have been reported. Differences may be due to the strain studied. A few representative mutant Envs were then tested for resistance. Based on the large panel of different mutations along with structural modeling, the authors conclude that there may be diverse mechanisms of enfuvirtide resistance.

Comments

The authors use their novel technology to present a set of experiments that are clearly presented, appropriately discussed, and likely to be of interest to those studying Env function and resistance to entry inhibitors. The strengths are the comprehensive panel of mutations studied and thoughtful discussion. Novel mutations identified may relate to the fact that literature on enfuvirtide resistance in clade A Envs are limited. A potential limitation inherent to the technology used is difficulties interpreting the significance of many of the identified novel mutations for the development of resistance. Some of the mutations identified might not be selected in vivo or in vitro if compensatory mutations could emerge during the selection; compensatory mutations may favor selection of a different profile of resistance mutations. Nonetheless, the findings provide new insights into potential modes of resistance.

Minor comments:

1)     Testing more mutant Envs for resistance would add value to the findings. Suggest including resistance data with statistics in the main paper rather than supplemental materials.

2)     A little more discussion about the DEAE dextran effect and plateau in the resistance curves from clonal pseudovirus stocks would be helpful.  

Author Response

Summary

The authors use their previously published methods involving deep mutational scanning to quantify the effect of all single amino-acid mutations in a clade A Env on resistance to enfuvirtide. They identified many mutations that were previously identified as resistance mutations in prior clinical and in vitro studies, as well as mutations that have not been previously identified. Most mutations clustered in the NHR, a region known to be important for enfuvirtide binding, but novel mutations in other regions of Env were also identified. The NHR mutations included some, but not all, of the most common enfuvirtide resistance mutations that have been reported. Differences may be due to the strain studied. A few representative mutant Envs were then tested for resistance. Based on the large panel of different mutations along with structural modeling, the authors conclude that there may be diverse mechanisms of enfuvirtide resistance.

Comments

The authors use their novel technology to present a set of experiments that are clearly presented, appropriately discussed, and likely to be of interest to those studying Env function and resistance to entry inhibitors. The strengths are the comprehensive panel of mutations studied and thoughtful discussion. Novel mutations identified may relate to the fact that literature on enfuvirtide resistance in clade A Envs are limited. A potential limitation inherent to the technology used is difficulties interpreting the significance of many of the identified novel mutations for the development of resistance. Some of the mutations identified might not be selected in vivo or in vitro if compensatory mutations could emerge during the selection; compensatory mutations may favor selection of a different profile of resistance mutations. Nonetheless, the findings provide new insights into potential modes of resistance.

Minor comments:

1)     Testing more mutant Envs for resistance would add value to the findings. Suggest including resistance data with statistics in the main paper rather than supplemental materials.

We have now moved one of the validation datasets, Supplemental Figure 1, into the main text as Figure 3, as was also suggested by Reviewer #4. We agree that validating additional resistance mutations will always add additional value to deep mutational scanning datasets. In antibody resistance studies in the past, we have validated many mutations (Dingens et al. Cell Host Microbe 2017, N=26 mutants for one antibody;  Dingens et al. Immunity 2019, N=16-19 mutants for three antibodies), and the results of our high-throughput assay have always been well-correlated with the results of testing individual mutation in traditional one-at-a-time assays. In the current study, we validate relatively few mutations. We chose a smaller set of mutations (and focused on novel mutations allosteric to the NHR binding site), as many of the mutations we mapped have been previously characterized as resistance mutations in detail by other studies. Additionally, the editor requested that we revise our manuscript based on reviewer comments in just 5 days, making additional experiments infeasible.

2)     A little more discussion about the DEAE dextran effect and plateau in the resistance curves from clonal pseudovirus stocks would be helpful. 

We agree that these are interesting but understudied aspects of enfuvirtide resistance. We have now expanded on our discussion of both the altered inhibition plateau and the effect of the DEAE-dextran concentration.

In addressing the altered inhibition plateau, we now state “This suggests that these mutations result in a sub-population of enfuvirtide resistant viruses, possibly by altering Env’s conformation or dynamics. While this is a well characterized mechanism of resistance for neutralizing antibodies [31,41], less data exists on incomplete enfuvirtide inhibition. However, it has been shown that enfuvirtide resistance mutations can alter the inhibition curve slope [37], which may be mechanistically related to the incomplete inhibition observed here. If this phenotype can manifest in vivo and a sub-population of viruses were resistant to enfuvirtide, this could present an opportunity for additional resistance mutations to arise.” (lines 262-269).

In addressing the effect of the DEAE-dextran concentration, which is closely related to the altered inhibition plateau, we now state “Indeed, the phenotype of some of the mutations we validated, particularly the mutations that resulted in incomplete inhibition in a cell culture inhibition assay, were dependent on the DEAE-dextran concentration. This could be related to the well-characterized phenomena that enfuvirtide sensitivity phenotypes commonly measured using cell culture-based experiments do not always correlate with enfuvirtide’s clinical efficacy [17,20,47].” (lines 287-291).

Reviewer 2 Report

This study represents a comprehensive analysis of amino acid substitutions in HIV-1 Env that confer resistance to enfuvirtide, a fusion inhibitor that is occasionally used in salvage antiretroviral therapy.  Authors have used a systematic approach (deep mutational scanning) to screen for mutations between codons 31 and 702 of Env, covering >12,000 possible amino acid substitutions. As expected authors confirm the relevance of several previously characterized resistance mutations (e.g. S553T, V549E or V549A) and identify others located away from the major enfuvirtide binding site.  The manuscript is well-written, experiments adequately performed and the paper is interesting for understanding antiretroviral drug resistance. 

Comments/questions that should be addressed in a revised manuscript:

1)      A few mutations in the IAS list are not selected in the study (e.g. G547D, G547S, I548V, Q550R,…).  Authors comment on V549M but not on the others.  Is this a limitation of their approach?  Are there any plausible explanations for these results?   Discussion or comments on these limitations should be included in the revised manuscript.

2)      Experiments have been carried out with a subtype A strain, when probably subtype B or C are more relevant.  Although I think that this is only a minor limitation, some justification should be included in the manuscript.

3)      Although “differential selection” has been previously described, I think that authors should add a few lines to describe in a few words what are the measurements they represent in ordinates in Fig. 1 and elsewhere in the text.

4)      Methods should not contain Results (delete references to File S1 and S2). This information should be provided in the Results section.

5)      The manuscript is generally well-written but needs some editing:

.- lines 40-41: sentence does not read well to me… please rewrite

.- Authors use ug/ml instead of µg/ml in many different places in the manuscript (lines 82, 84, 113, supplemental Figs. 3 and 4A,B, etc…).  Please correct and leave a space between numbers and units.

.- Typos: pseudoviruses (line 102), commas and spaces in ref. 28

.- Supplemental Figure 2 should be provided at a larger size.

Author Response

Comments and Suggestions for Authors

This study represents a comprehensive analysis of amino acid substitutions in HIV-1 Env that confer resistance to enfuvirtide, a fusion inhibitor that is occasionally used in salvage antiretroviral therapy.  Authors have used a systematic approach (deep mutational scanning) to screen for mutations between codons 31 and 702 of Env, covering >12,000 possible amino acid substitutions. As expected authors confirm the relevance of several previously characterized resistance mutations (e.g. S553T, V549E or V549A) and identify others located away from the major enfuvirtide binding site.  The manuscript is well-written, experiments adequately performed and the paper is interesting for understanding antiretroviral drug resistance. 

Comments/questions that should be addressed in a revised manuscript:

1)      A few mutations in the IAS list are not selected in the study (e.g. G547D, G547S, I548V, Q550R,…).  Authors comment on V549M but not on the others.  Is this a limitation of their approach?  Are there any plausible explanations for these results?   Discussion or comments on these limitations should be included in the revised manuscript.

Not identifying a number of previously identified resistance mutations is certainly a limitation of our approach. We now include an extended discussion point on this limitation, and added Supplemental Figure 4 to show how this limitation may be driven by the differential mutational tolerance of the subtype A strain we used here as compared to subtype B strains commonly used in prior work on enfuvirtide resistance. We also comment specifically on additional previously characterized resistance mutations that we did not identify in our experiments, stating “For example, we do not identify the G547D and G547S resistance mutations, which were identified by enfuvirtide selection of lab-adapted, subtype B viruses in cell culture [24]. Previous studies show that these mutations are tolerated in a lab-adapted subtype B virus (LAI), but they are not well-tolerated by the subtype A variant BG505 during viral replication in cell culture (Supplemental Figure 4) [34,35]. Similarly, the known resistance mutation I548V, which was not enriched in our experiments, is better tolerated in LAI than in BG505 (Supplemental Figure 4). These differences highlight how the balance between enfuvirtide resistance and replicative fitness may depend on the virus strain.” (lines 274-286).

2)      Experiments have been carried out with a subtype A strain, when probably subtype B or C are more relevant.  Although I think that this is only a minor limitation, some justification should be included in the manuscript.

We now address this point in the text, stating “We used this strain because relatively little is known about enfuvirtide resistance in subtype A viruses, and because BG505 is widely used in structural studies [34,35].” (lines 78-79).

3)      Although “differential selection” has been previously described, I think that authors should add a few lines to describe in a few words what are the measurements they represent in ordinates in Fig. 1 and elsewhere in the text.

We thank the reviewer for pointing out an additional way to make our manuscript more easily understood. We now describe the differential selection statistic in both the Analysis of deep sequencing data section of the Material and Methods, as well as in the legend to Figure 1.

4)      Methods should not contain Results (delete references to File S1 and S2). This information should be provided in the Results section.

We have now referenced Supplemental Files 1 and 2 in relevant portion of the Results section. However, we have also kept these in the Data availability and source code section of the Materials and Methods; we feel that it is critical to mention the computational analysis (File S1) and the raw results (File S2) in this section to facilitate the easy re-analysis of these datasets.

5)      The manuscript is generally well-written but needs some editing:

.- lines 40-41: sentence does not read well to me… please rewrite

.- Authors use ug/ml instead of µg/ml in many different places in the manuscript (lines 82, 84, 113, supplemental Figs. 3 and 4A,B, etc…).  Please correct and leave a space between numbers and units.

.- Typos: pseudoviruses (line 102), commas and spaces in ref. 28

.- Supplemental Figure 2 should be provided at a larger size.

We thank the reviewer for identifying these typos and opportunities to improve the manuscript. We have made each of these changes

Reviewer 3 Report

Description: The authors used a deep mutational scanning approach to identify mutations in the envelope gene of clade A BG505 that confer resistance to enfuvirtide (T20).  Independent Env mutant libraries were generated in which amino acids 31-702 were substituted with all possible substitutions. Viruses produced from the mutant libraries were used to infect SupT1.CCR5 cells in the presence or absence of enfuvirtide and functional viruses were selected and subsequently characterized by deep sequencing.  The results revealed many previously identified as well as novel resistance mutations, suggesting alternative pathways to acquiring resistance. 

General critique: This manuscript is well written and provides many resistance mutations; the degree of resistance could be quantified using the deep sequencing data. The results will be of interest to HIV researchers who are using T20/enfuvirtide like peptides as research tools.  Although this approach has apparently been previously used by these authors for other studies, the studies highlight the utility of using this approach to understanding drug resistance mechanisms.  As the authors discussed, there are many caveats to the approach; many of the resistance mutations may be unique to the experimental design, since different concentrations of DEAE-Dextran seem to make a difference in the level of resistance for some of the identified mutations. 

Overall the paper should be of interest to HIV-1 researchers interested in HIV-1 envelope and drug resistance studies as well as those interested in possibly applying this approach to other viral genes and inhibitors to elucidate drug resistance pathways.

Author Response

Comments and Suggestions for Authors

Description: The authors used a deep mutational scanning approach to identify mutations in the envelope gene of clade A BG505 that confer resistance to enfuvirtide (T20).  Independent Env mutant libraries were generated in which amino acids 31-702 were substituted with all possible substitutions. Viruses produced from the mutant libraries were used to infect SupT1.CCR5 cells in the presence or absence of enfuvirtide and functional viruses were selected and subsequently characterized by deep sequencing.  The results revealed many previously identified as well as novel resistance mutations, suggesting alternative pathways to acquiring resistance. 

General critique: This manuscript is well written and provides many resistance mutations; the degree of resistance could be quantified using the deep sequencing data. The results will be of interest to HIV researchers who are using T20/enfuvirtide like peptides as research tools.  Although this approach has apparently been previously used by these authors for other studies, the studies highlight the utility of using this approach to understanding drug resistance mechanisms.  As the authors discussed, there are many caveats to the approach; many of the resistance mutations may be unique to the experimental design, since different concentrations of DEAE-Dextran seem to make a difference in the level of resistance for some of the identified mutations. 

Overall the paper should be of interest to HIV-1 researchers interested in HIV-1 envelope and drug resistance studies as well as those interested in possibly applying this approach to other viral genes and inhibitors to elucidate drug resistance pathways.

We thank the reviewer for the well-summarized and detailed review of our manuscript.

Reviewer 4 Report

In the manuscript by Dingens et al the authors studied the effect of single amino acid mutations of clade A Env on resistance to enfurvirtide. The authors identify some previously characterized and some novel resistance mutations in the NHR that likely act during different fusion steps by altering fusion kinetics or exposure of the enfuvirtide binding site. The manuscript is heavily built on existing data and adds some novel information to the field of HIV resistance. Listed are a few specific concerns:

The manuscript would benefit by inclusion of a Table describing the Enfuvirtide resistance mutations identified in clade A, B, C, D and circulating recombinant forms. Please also include the region of Envelope the mutations are identified in and the advantage they confer in that particular clade.

How do the resistant mutants identified in the study fare in infectivity when compared to WT virus? Are they more fit or less than WT? It would be relevant to provide infectivity data in both TZM cells and SupT1.CCR5 cells.

Have the authors tested the resistant mutants for inhibition against other entry inhibitors? The most potent mutant like V549E should be tested against other inhibitors.

Are any of the mutations identified in the study found in naturally occurring HIV isolates? If yes, what would be the advantage of such mutations?

The mutant DNA libraries were passaged in SupT1.CCR5 cells. However, the validation of enfuvirtide resistant mutants was conducted in TZM-bl cells (Supplemental Figure 3 and 4). It would be more reasonable if the inhibition curves were also generated in SupT1.CCR5 cells.

Minor: Please move supplemental Figure 3 to the main text as provides important information. Also, please remove one WT curve from supplemental Figure 3A (Q552R).

Author Response

Comments and Suggestions for Authors

In the manuscript by Dingens et al the authors studied the effect of single amino acid mutations of clade A Env on resistance to enfurvirtide. The authors identify some previously characterized and some novel resistance mutations in the NHR that likely act during different fusion steps by altering fusion kinetics or exposure of the enfuvirtide binding site. The manuscript is heavily built on existing data and adds some novel information to the field of HIV resistance. Listed are a few specific concerns:

The manuscript would benefit by inclusion of a Table describing the Enfuvirtide resistance mutations identified in clade A, B, C, D and circulating recombinant forms. Please also include the region of Envelope the mutations are identified in and the advantage they confer in that particular clade.

This is an astute point, and Reviewer 2 made related suggestions. Most prior studies on enfuvirtide resistance have focused on subtype B strains. There are a handful of studies that examine sequence conservation within the enfuvirtide binding site of non-subtype B strains, and fewer studies that examine enfuvirtide sensitivity of non-subtype B strains. However, to our knowledge, there is very limited data on resistance mutations to enfuvirtide in non subtype B strains. As such, the suggested table on differences in resistance mutations between strains/subtypes would be interesting, but it would be very sparse based on existing data. Therefore, in lieu of the table, we have described the prior studies on enfuvirtide sensitivity across subtypes in the text, stating, “Much of the prior work on enfuvirtide sensitivity and resistance has focused on subtype B viruses [17,18]. Sequence variability within enfuvirtide’s binding site differs across subtypes, but a limited number of studies have suggested there are not substantial differences in enfuvirtide sensitivity or resistance mutations between subtypes [29–33].” (lines 68-71).

How do the resistant mutants identified in the study fare in infectivity when compared to WT virus? Are they more fit or less than WT? It would be relevant to provide infectivity data in both TZM cells and SupT1.CCR5 cells.

This is an excellent question, and we thank the reviewer for providing the opportunity to strengthen our manuscript by including this comparison. In previous work, we have used deep mutational scanning in the absence of enfuvirtide selection to quantify the effect of all mutations to Env on just viral replication in SupT1.CCR5 cells (Haddox et al. PLoS Pathogens 2016; Haddox et al. eLife 2018). This provides a dataset related to the cell culture infectivity/fitness of each mutant we identify in our resistance profiling. We have included an additional figure (Supplemental Figure 4) making this comparison. We discuss these data in the text in the context of potential subtype-specific fitness effects, stating “For example, we do not identify the G547D and G547S resistance mutations, which were identified by enfuvirtide selection of lab-adapted, subtype B viruses in cell culture [24]. Our previous studies show that these mutations are tolerated in a lab-adapted subtype B virus (LAI), but they are not well-tolerated by the subtype A variant BG505 during viral replication in cell culture (Supplemental Figure 4) [34,35]. Similarly, the known resistance mutations I548V, which was not enriched in our experiments, is better tolerated in LAI than in BG505 (Supplemental Figure 4). These differences highlight how the balance between enfuvirtide resistance and replicative fitness may depend on the virus strain.” (lines 274-286).

However, we have not included infectivity in TZM-bl cells; we do not feel this is as relevant as the SupT1.CCR5 data we added. Additionally, the editor requested that we revise our manuscript based on reviewer comments in just 5 days, making additional experiments infeasible.

Have the authors tested the resistant mutants for inhibition against other entry inhibitors? The most potent mutant like V549E should be tested against other inhibitors.

We agree that it is of interest to examine if the resistance mutations mapped here also results in resistance to additional fusion inhibitors. However, we feel that this is beyond the scope of this manuscript, which focuses on the complex mechanisms of enfuvirtide resistance and highlighting the use of deep mutational scanning to profile drug resistance. Additionally, as stated above, the editor requested that we revise our manuscript based on reviewer comments in just 5 days, making additional experiments infeasible. We do, however, thank the reviewer for suggesting this additional line of future work.

Are any of the mutations identified in the study found in naturally occurring HIV isolates? If yes, what would be the advantage of such mutations?

This is another excellent point to consider. We have now compared the resistance profiling to the amino acid frequencies found in nature in the added Supplemental Figure 4.  Generally, most resistance mutations we (and others) have identified are not or are rarely found in nature, suggesting that they have a fitness impact, which is another point also addressed in Supplemental Figure 4.

The mutant DNA libraries were passaged in SupT1.CCR5 cells. However, the validation of enfuvirtide resistant mutants was conducted in TZM-bl cells (Supplemental Figure 3 and 4). It would be more reasonable if the inhibition curves were also generated in SupT1.CCR5 cells.

We agree that using SupT1.CCR5 cells for validation would most closely mimic the experimental conditions used in the deep mutational scanning experiments. However, we used pseudovirus entry into TZM-bl cells for validation as this is a highly standardized and commonly used reporter cell line assay in which we can easily generate inhibition curves. Additionally, using an alternative cell line helps to generalize the results found in SupT1.CCR5 cells to other cell culture systems.

Minor: Please move supplemental Figure 3 to the main text as provides important information. Also, please remove one WT curve from supplemental Figure 3A (Q552R).

We have now moved Supplemental Figure 3 to the main text as Figure 3, and we have removed the duplicate WT curves (previously included to show there is very little noise between replicates).